# Living Bacteria: A New Vehicle for Vaccine Delivery in Cancer Immunotherapy

**DOI:** 10.3390/ijms26052056

**Published:** 2025-02-26

**Authors:** Min Yang, Peiluan Zhong, Pengcheng Wei

**Affiliations:** Guangxi Key Laboratory of Special Biomedicine, School of Medicine, Guangxi University, Nanning 530004, China; minyyang_93@outlook.com (M.Y.); 2328391063@st.gxu.edu.cn (P.Z.)

**Keywords:** cancer vaccine, vaccine delivery, bacteria, immunotherapy

## Abstract

Cancer vaccines, aimed at evolving the human immune system to eliminate tumor cells, have long been explored as a method of cancer treatment with significant clinical potential. Traditional delivery systems face significant challenges in directly targeting tumor cells and delivering adequate amounts of antigen due to the hostile tumor microenvironment. Emerging evidence suggests that certain bacteria naturally home in on tumors and modulate antitumor immunity, making bacterial vectors a promising vehicle for precision cancer vaccines. Live bacterial vehicles offer several advantages, including tumor colonization, precise drug delivery, and immune stimulation, making them a compelling option for cancer immunotherapy. In this review, we explore the mechanisms of action behind living bacteria-based vaccines, recent progress in popular bacterial chassis, and strategies for specific payload delivery and biocontainment to ensure safety. These approaches will lay the foundation for developing an affordable, widely applicable cancer vaccine delivery system. This review also discusses the challenges and future opportunities in harnessing bacterial-based vaccines for enhanced therapeutic outcomes in cancer treatment.

## 1. Introduction

Cancer continues to be one of the most urgent global public health challenges, with its incidence and mortality rates steadily increasing [1,2]. Despite advancements in conventional therapies such as surgery, radiation, and chemotherapy, cancer patients often experience poor outcomes due to recurrence, metastasis, and the toxicities of treatment [3,4]. Immunotherapy has emerged as a transformative approach in cancer treatment, harnessing the body’s immune system to target and eliminate established tumors, and is now considered one of the most promising therapeutic strategies [5,6]. Extensive research on novel cancer immunotherapies, including immune checkpoint inhibitors (ICIs), cancer vaccines, and chimeric antigen receptor (CAR)-T and T cell receptor (TCR) therapies, have led to the clinical approval of several innovative treatments [7,8,9,10]. However, the clinical utility of cancer immunotherapies is often limited by factors such as poor patient response, life-threatening off-target side effects, long treatment timelines, and high costs [11,12,13,14]. For example, a range of inflammatory adverse effects, or immune-related adverse events (irAEs), can arise from the disinhibition of T cell function by ICIs [15]. Furthermore, despite 43% of patients qualifying for ICI treatment, only 12% actually benefit from it [16]. While CAR-T therapy has shown significant success in treating blood cancers, it is associated with potential side effects, including cytokine release syndrome and neurotoxicity [11,17]. Additionally, concerns have arisen about the possibility of secondary malignancies, as genetic modifications in T cells may increase the risk of tumorigenesis due to off-target effects or insertional mutagenesis [18,19,20,21].

Recently the emerging cancer vaccines have garnered renewed attention, particularly due to advances in the study of tumor-specific antigens. Cancer vaccine, including nucleic acid vaccines (DNA and mRNA), peptide and protein vaccines, and cell-based vaccines, are designed to prime antigen-specific T cells and elicit a potent immune response against cancer [10,22,23,24,25]. Early efforts to apply vaccines in oncology can be traced back to the work of William Coley in the early 20th century, who pioneered the use of bacterial injections to treat cancers [26]. In the 1950s, the introduction of *Bacillus Calmette-Guérin* (BCG) by Lloyd Old further cemented the potential of immunotherapy in cancer treatment [27]. Fast forward to the late 20th century, when the identification of tumor-specific antigens in the 1990s opened new avenues for precision cancer vaccines [28]. The approval of Sipuleucel-T in 2010 marked a key milestone, confirming the clinical relevance of cancer vaccines and reigniting the interest in their development [29]. The advancements in vaccine technology driven by the COVID-19 pandemic have rekindled interest in their potential applications in oncology, further enhancing the prospects of cancer vaccines [30].

Though cancer vaccines have suggested easy operation and effectiveness, there are still several challenges, including poor targeting of the delivery system, vaccine stability and safety, insufficient immunogenicity, and the immunosuppressive tumor microenvironment [9]. To solve some key problems, various encapsulation and delivery platforms have been developed, including nanoparticles, liposomes, and outer membrane vesicles (OMVs) [31]. These platforms provide enhanced stability, targeted delivery, and the potential for controlled release, making them key tools in overcoming the limitations of traditional vaccine formulations. Ferritin, as a representative nanoparticle platform, has shown considerable potential as an antigen-presenting vehicle, successfully generating robust antibody responses against various pathogens, including SARS-CoV-2 [32], Epstein–Barr virus [33], influenza A [34], and respiratory syncytial virus [35]. Recently, ferritin-based nanoparticles have emerged as a promising delivery system for neoantigen peptides, as they efficiently drain to lymph nodes and stimulate potent immune responses against tumors [36]. For mRNA vaccines, liposome-based platforms have gained significant attention due to their ability to encapsulate mRNA efficiently, protecting it from enzymatic degradation [37]. In cancer immunotherapy, FixVac, a systemic mRNA vaccine encoding four non-mutated melanoma antigens based on a liposome platform, has shown promising results in a Phase I trial (NCT02410733), where immune responses against at least one of the antigens were triggered in over 75% of patients [38]. Similarly, OMV-based vaccines have gained recognition for their success in combating *Neisseria meningitidis B* in New Zealand [39]. OMVs contain key antigenic components that elicit protective immune responses, making them an ideal adjuvant for priming immunity [40]. However, despite this progress, existing vaccine delivery systems face significant challenges in directly targeting tumor cells and delivering adequate amounts of antigen [9]. Therefore, the search for more efficient and targeted vaccine delivery systems—such as those utilizing live bacteria—has become a focal point, offering a promising new frontier in the battle against cancer [41]. Living bacterial vehicles are particularly promising due to their inherent ability to colonize tumors [42]. In addition, bacteria can be genetically engineered to deliver drugs with high precision, enabling the accumulation of therapeutic agents at targeted tumor sites while minimizing the risk of systemic toxicity. Moreover, bacteria and their derivatives have the ability to stimulate antitumor immunity [43,44], thereby acting as potent adjuvants to initiate a robust immune response. Given these advantages, living bacteria—first utilized by William Coley in 1891 for cancer treatment—have once again captured the attention of researchers for their potential in tumor-targeted delivery of various compounds in cancer immunotherapy.

In this review, we present a comprehensive exploration of the diverse strategies for developing vaccines based on living bacteria. We begin by outlining how such vaccines leverage live bacteria to activate the immune system and target tumor destruction. Next, we discuss commonly used bacterial chassis that serve as delivery systems for these vaccines. Additionally, we provide an overview of the latest advancements in payload release techniques and conclude with a forward-looking perspective on the challenges and opportunities in this promising new approach to cancer immunotherapy delivery.

## 2. Mechanisms of Action in Living Bacteria-Based Vaccines

To kill tumors, APCs must efficiently receive signals from tumor antigens, followed by activation of various immune cells, including CD8+ cytotoxic T cells (CTLs) and CD4+ helper T cells [45]. To achieve this goal, first, various cancer vaccines are carefully designed and selected, including DNA, RNA, protein, peptide-based vaccines, and cell-based vaccines [10]. Adjuvants are often used in conjunction with vaccines to enhance the speed and strength of this immune response by activating innate immune pathways [46]. The selection and design of tumor antigens are crucial, as the ability to elicit a specific T cell response is directly determined by the chosen antigens [47]. Tumor antigens are molecules expressed by cancer cells that are recognized by the immune system as foreign or abnormal. These antigens are categorized based on their origin, immunogenicity, and their expression in both tumor and normal tissues. Broadly, tumor antigens can be classified into two major categories: tumor-specific antigens (TSAs) and tumor-associated antigens (TAAs). TSAs are unique to tumor cells and are not expressed in normal tissues. They typically arise from mutations in the cancer genome, leading to the production of novel peptides that are presented on the tumor cell surface in association with Major Histocompatibility Complex (MHC) molecules, also referred to as Human Leukocyte Antigen (HLA) in humans. These antigens are highly specific to the tumor and are considered ideal targets for immunotherapy. A prominent example of tumor-specific antigens (TSAs) is neoantigens, which are primarily generated by somatic mutations but can also arise from other genetic alterations and are exclusively present in the tumor, making them highly specific to individual cancers [48,49]. These neoantigens show particular promise in solid tumors, where they offer a unique opportunity for targeted therapeutic interventions [50,51]. TAAs are expressed in both tumor cells and normal tissues, with a significantly elevated expression in cancer cells. Some well-known examples of TAAs include human epidermal growth factor receptor 2 (HER2/neu), melanoma-associated antigen 1 (MART-1), and carcinoembryonic antigen (CEA) [52]. Although TAAs can be targeted for immunotherapy, their potential for causing off-target effects due to their presence in normal tissues remains a key challenge [53].

Next, the delivery system is engineered to load the cargo. Given that distinct tumor microbiomes exist in different tumor types and selectively colonize immune-privileged tumor cores [54,55], bacteria have a unique advantage in consistently and precisely delivering a wide range of drug types. After the vaccine is taken up, the loaded bacteria then spontaneously target and colonize the tumor tissue (Figure 1a). Due to the complex tumor microenvironment, the bacteria proliferate and express the drug. Through genetic control mechanisms, including inducible genetic circuits, quorum sensing, and tumor sensing, the payload is triggered to be expressed in large quantities at the tumor site [56]. Subsequently, through mechanisms such as lysis systems or surface presentation, the drug may be internalized into phagosomes by APCs, or it may directly act extracellularly by killing the tumor or aiding immune cells. In the case of a prolonged inflammatory immune response, immature DCs will undergo maturation, processing the drug while losing their ability to absorb antigens [57]. After maturation, the peptide is presented by MHC molecules, and recognized by T cells, activating CD4+ (helper) and/or CD8+ (cytotoxic) T cells (CTLs) (Figure 1b) [58]. CD4+ T helper cells enhance the immune response by activating CTLs through the secretion of interleukin-2 (IL-2) and interferon-gamma (IFN-γ). In addition, DCs secrete IL-12 to enhance the production of costimulatory factors [59]. After identifying the antigens on tumor cells, CTLs induce apoptosis in the tumor cells, initiating a cellular immune response. Upon recognition of tumor-specific antigens, CD8+ cytotoxic T lymphocytes (CTLs) initiate apoptosis in tumor cells, thereby triggering a cellular immune response (Figure 1c) [60]. Concurrently, activated CD4+ T helper cells enhance the immune response by promoting the phagocytic activity of macrophages, augmenting the cytotoxic capabilities of natural killer (NK) cells, and stimulating B cells [60,61]. This cascade of immune activation facilitates the differentiation of B cells into plasma cells, which produce antigen-specific antibodies, thereby contributing to a humoral immune response. These antibodies can bind to and neutralize tumor-associated antigens, facilitating tumor clearance through mechanisms such as antibody-dependent cellular cytotoxicity (ADCC) and complement activation (Figure 1c) [62]. Moreover, cytokines and antibodies secreted by immune cells function not only in immune modulation but also as potent adjuvants [63]. When co-delivered with peptide or genetic vaccines via bacterial vectors, they further enhance the immune response. The ability of bacteria to deliver these immune mediators in situ maximizes both the efficacy and speed of the immune activation, thereby promoting a more robust and targeted antitumor response.

## 3. Bacterial Chassis

Bacteria, due to their ability to preferentially grow in the hypoxic and necrotic regions of the tumor microenvironment, represent a promising vehicle for delivering a variety of compounds directly to tumors, thereby inducing an immune response in situ [64]. Several obligate or facultative anaerobic bacteria, such as *Salmonella*, *Lactobacillus*, *Listeria*, and *E. coli*, have demonstrated the ability to colonize tumors.

*Bacille Calmette Guérin* (*BCG*), an attenuated strain of *Mycobacterium bovis*, has long been used clinically in the treatment of non-muscle invasive bladder cancer (NCT02015104), serving as a benchmark for FDA-approved bacterial cancer therapies. However, its efficacy decreases when the cancer has invaded deeper layers of the bladder wall, particularly the muscular layer. This limitation stems from the fact that BCG primarily elicits a localized immune response, and its effectiveness is often dependent on the ability to reach and stimulate immune cells in the tumor, which becomes more difficult as the tumor penetrates deeper layers. Designing such a vaccine for muscle-invasive bladder cancer requires much energy to finely arrange the selection of payload, the bacteria strain, delivery method, genetic controlled payload, and good colonization. Some novel strategies enhance the efficacy of bacterial therapy for muscle-invasive bladder cancer. Maybe some combination of therapies is well suited to these complex cancers. Beyond *BCG*, both attenuated bacterial strains and genetically engineered strains with therapeutic payloads are currently undergoing clinical evaluation in cancer patients (Table 1) [65].

*Salmonella typhimurium* is one of the most extensively studied bacterial species in the context of bacterial cancer therapy and has been employed in numerous clinical trials [65]. In addition to its role in cancer therapy, *Salmonella* is a significant zoonotic pathogen, frequently responsible for causing salmonellosis in both humans and animals. This infection is commonly characterized by symptoms such as diarrhea and sepsis. Notably, attenuated *Salmonella* strains are more effective than inactivated vaccines in inducing humoral, cellular, and mucosal immunity, making them a promising candidate for vaccine development [66,67]. As a versatile vector, it also serves as an efficient carrier for foreign genes and a natural mucosal immune adjuvant, which has contributed to its widespread recognition and application in the medical community [68]. A unique feature of *Salmonella* is its Salmonella pathogenicity island (SPI)-based type three secretion systems (T3SS). Both SPI-1 T3SS and SPI-2 T3SS exhibit remarkably tight regulation in the timing of their expression and the spatial localization of the encoded effectors during the infection process within host cells. SPI-1 T3SS primarily manipulates adhesion to the host cell membrane, while SPI-2 T3SS injects effector proteins into the host cells, facilitating further manipulation of host cell functions [69,70]. Since *Salmonella* infects various types of cells, including both phagocytic and non-phagocytic cells, it has the potential to deliver effectors to both intracellular and extracellular spaces [44]. However, since *Salmonella* is pathogenic, it must exhibit a strong safety profile to serve as a vaccine vector. Therefore, effective attenuation of *Salmonella* is a primary requirement for its use as a vaccine vector. Genetic engineering has led to the development of several attenuated *Salmonella* strains as candidates for cancer therapy, including VNP20009, which has deletions in *msbB* and *purI*, as well as strains with *△ppGpp* and A1-R modifications [71,72,73].

*Listeria monocytogenes* primarily affects the gut and spreads through the lymphatic system. In both mouse and human trials, genetically modified *Listeria* have proven effective as vectors for delivering antigen to tumors [74,75,76]. By deleting genes such as *actA*, *inlA*, and *inlB*, which prevent intercellular transport and epithelial cell infection, these modified strains become safe while concentrating their delivery to antigen-presenting cells (APCs) [77]. Once internalized by APCs, the native LLO expressed by *Listeria* help lyse the bacteria membrane and vesicle membranes, facilitating the translocation of the payload into the cytosol.

*Clostridium novyi* (*C. novyi*) is a spore-forming anaerobe that only germinates in low-oxygen environments, such as those found in tumor necrosis or the gut lumen, which keeps healthy tissue intact [78]. Deletion of the alpha toxin makes it a safe therapeutic vehicle [78]. Single injections of non-toxic *C. novyi* produced immunological responses and showed antitumor activity in an early clinical trial (NCT01924689).

The gut commensal strains *E. coli* MG1655 and *E. coli* Nissle 1917 (EcN) are emerging as popular chassis for therapeutic applications due to their presence in the intestinal microbiome and good adaptability to colonize tumors. These bacteria are capable of thriving in the immunosuppressive and biochemically distinct microenvironment induced by the pathological alterations associated with solid tumors, supported by their chemotactic properties that direct them to tumor sites [79]. Given its established human safety record, generally recognized as safe (GRAS) status, genetic tractability, and the expanding tools of synthetic biology, *E. coli* has become a popular platform for creating intelligent microorganisms [80,81]. Recently, there has been a surge of research focused on employing engineered *E. coli* for cancer immunotherapy. For instance, SYNB1891 is engineered to deliver STING agonists directly to tumors, activating the STING pathway. This triggers an immune response by promoting type I interferon production, stimulating dendritic cells, NK cells, and CTLs to enhance anti-tumor immunity. It is currently undergoing clinical trials to assess its effectiveness in cancer treatment [82,83]. Li et al. used engineered EcN for the local quorum-regulated delivery of interferon-γ [84]. Redenti et al. incorporated several strategies into EcN to deliver massive neopeptide-containing peptide arrays to the cytosol. In this approach, EcN was engineered to express peptide arrays that contain both CD4+ T cell and CD8+ T cell epitopes derived from neoantigens, which are generated by somatic mutations in tumor cells [80]. Yang et al. engineered *DH5α* to display a decoy-resistant IL-18 mutein on its surface for targeted delivery to tumors. IL-18 is a cytokine that activates immune cells like natural killer (NK) cells and CD8+ T cells. The decoy-resistant mutein bypasses inhibitory receptors, enhancing IL-18’s immune-boosting effects [81]. All of these immunotherapies have achieved significant therapeutic results in animal models by activating CD4+, CD8+, and NK cells to suppress tumor progression. Meanwhile, several variants have been developed for cancer treatment, including SYNB1891, SLIC variants, EdaH1-HlpA, and L-arginine EcN [85].

Recently, *Clostridium butyricum*, which produces high levels of tryptophan, has been found to selectively colonize tumors and inhibit the activity of indoleamine 2,3-dioxygenase (IDO) by generating butyrate. This enhances the tumor microenvironment and reprograms the CD8+ T cell response within tumors [86]. *Staphylococcus epidermidis*, as part of the skin microbiome, can induce a highly specific adaptive immune response. It has been genetically engineered to express melanoma tumor antigens, leading to the production of tumor-specific T cells that target and kill the melanoma cells [87]. Furthermore, intestinal microbes have been shown to generate metabolites that promote carcinogenesis and influence the effectiveness of immunological treatments [88]. Microbiomes present in various tumor sites can both promote and inhibit cancer cell growth [89,90]. These strains could offer a new avenue for combined cancer immunotherapy. As research progresses, additional strains are being developed as potential new vehicles.

## 4. Vaccine Delivery

Effective molecular delivery is a critical element for the success of bacterial vaccines, especially for therapeutic applications. Since therapeutic drugs are usually expressed inside the bacteria and are not readily accessible to APCs, several elegant delivery systems have been engineered to release the payload. There are three predominant delivery systems: intracellular delivery with externally triggered lysis, bacteria surface delivery, and protein secretion (Table 2).

### 4.1. Intracellular Delivery and Lysis System

A major focus in the current research is encapsulating vaccines inside bacteria. However, except for *Salmonella* and *Yersinia enterocolitica*, which generate T3SS injectisomes to deliver therapeutic proteins directly into the cancer cells’ cytoplasm in a rapid and controlled manner [103], most bacteria cannot release the payload into the cells. Therefore, a release system is essential for intracellular drug delivery. Below, we summarize several release mechanisms and their applications in bacteria. The *PsseJ-lysE* in VNP20009, which couples the *Salmonella* promoter *PsseJ* with a suicide gene (*lysE* from phage phi X174), is used to sense the vacuolar environment and trigger lysis after invasion into cancer cells. *PsseJ*, one promoter of the SPI2 gene, is activated after invasion into cancer cells. The re-expressed flhDC, which is created by deletion of flhDC in the genome while adding a controllable re-expression, can enhance the invasion efficiency. Once VNP20009 invades the cell, *lysE* is expressed under the activation of *PsseJ*, followed by bacterial lysis and the drug releasing into the cytoplasm (Figure 2a) [91].

The synchronized lysing circuit (SLC) enables repeated release of genetically encoded cargo in situ through synchronized lysis triggered by a threshold population density [104,105]. This ensures that cargo is only released in areas such as tumors, where bacteria can proliferate to a sufficiently high density. The circuit is controlled by the *luxI* promoter and three genes, including two activators (autoinducer (AHL), *LuxR*) and *lysE* (Figure 2b). The expression of all three genes is controlled by *luxI*. Specifically, the *luxI* promoter regulates the synthesis of AHL, which binds to *LuxR*, enabling the promoter to be transcriptionally activated. Meanwhile, *lysE*, also regulated by the *luxI* promoter, leads to negative feedback and ultimately results in cell death. Crucially, AHL provides an intercellular synchronization mechanism by diffusing to neighboring cells [104]. This system is widely used in the probiotic EcN strain for the localized release of drugs in cancer immunotherapies. Nanobodies targeting programmed cell death protein ligand-1 (PD-L1) and cytotoxic T lymphocyte-associated protein-4, when combined with a single injection of the engineered lysis system, demonstrated an improved therapeutic response, including activation of tumor-infiltrating T cells and rapid tumor regression, compared to similar clinically relevant antibodies alone [94]. Similar immune responses have been observed with human chemokine CXCL16 [93] and a nanobody antagonist of CD47 [92]. IFN-γ, in conjunction with the PD-1 blockade, has shown great potential as a therapeutic approach to improve overall antitumor immunity across a range of solid tumor types [108,109]. However, due to the widespread distribution of IFN-γ receptors, severe dose-limiting toxicities have hindered its use [110,111]. Recently, this dose limitation was addressed by Li et al. through the use of EcN combined with the SLC system [84].

Listeriolysin O (LLO) can facilitate the delivery of co-expressed proteins to the macrophage cytosol in bacterial vaccines. LLO is a pore-forming cytolysin secreted by various species of Gram-positive bacteria [96]. It was previously used in liposomal vaccines to deliver co-encapsulated proteins into the cytosol of macrophages [112]. After homing in on the tumor, the bacterial vaccine is engulfed by macrophages, where it ends up in phagosomes. LLO creates pores in the bacterial membrane, causing bacterial lysis and releasing LLO and the target protein into the phagosome. This leads to the formation of large pores in the phagosomal membrane, allowing the drug to translocate into the cytosol (Figure 2c) [106,113]. In theory, any drug that *E. coli* can synthesize and is co-expressed with LLO can be delivered into the cytosol of macrophages. Recently, Andrew et al. engineered EcN with LLO to deliver neoepitope-containing peptide arrays for antitumor vaccination, resulting in specific, effective, and durable systemic antitumor immunity [80]. Meanwhile, LLO is also well-suited for delivering short hairpin RNA (shRNA), effectively inducing gene silencing for cancer gene therapy [95,114].

Both SLC and LLO require the introduction of additional plasmids into bacteria, which could compete for the translation machinery or raise concerns about immunogenicity. To address this issue, Liu et al. engineered EcN with deletion of the *lpp* gene (Δlpp), which encodes the lipoprotein responsible for covalently attaching the bacterial outer membrane. This deletion weakens the outer membrane, allowing for the direct release of expressed biologics (Figure 2d) [97,115].

### 4.2. Surface Delivery and Surface Presentation

Vaccines that display antigens or antibodies on the surface of bacteria can elicit both humoral and cell-mediated immunity, which together provide a more robust immune response than either type alone [116]. There are two main approaches to surface display: fusion proteins and the SpyCatcher-SpyTag (SC-ST) system.

Proteins and peptides can be fused with anchoring motifs and displayed on the surface of microbial cells. Small proteins and peptides can be surface exposed using various carrier proteins, including *E. coli* protein MipA, lLPxTG, Omp, PsgA, N-terminal transmembrane helices, and S-layer proteins [117,118,119,120]. For example, HPV E7, fused to PgsA and displayed on the surface of *Lacticaseibacillus casei*, elicits a stronger immune response from cytotoxic T lymphocytes than when E7 is secreted or intracellularly accumulated in *L. lactis* [98], and this is currently being tested in a Phase 2 clinical trial. Yang et al. first explored the surface display of key immune-activating cytokines by fusing them with *E. coli* outer membrane protein A, the C-terminal of IgA proteinase, the N-terminal of *eaeA*, and *YiaT* (Figure 2e) [81]. In contrast, large and complex proteins are better displayed using autotransporter proteins or ice nucleation proteins (INPs) [117]. INPs, with their unique structural and functional properties, have proven to be an excellent carrier for displaying large and intricate proteins [121]. For example, MAP3061c was successfully displayed on the *E. coli* cell surface using an INP surface display system [99].

Due to limitations in the fusion protein method, such as the fusion site and protein size, the SpyCatcher-SpyTag (SC-ST) bioconjugation system was developed to offer a wider range of physiological compatibility [107]. Correctly folded proteins that are independently expressed as genetic fusions with SC and ST domains can bind covalently, forming an isopeptide bond between SC and ST. Membrane anchor Lpp-OmpA genetically fuses to the SC domain, and the payload is fused to the ST. With covalent interaction formed by post-translational SC-ST, the ligated complex enters into the outer membrane through the Lpp-OmpA, thus successfully displaying the drug on the cell surface (Figure 2f).

### 4.3. Secretion Delivery

One of the simplest and most popular methods for delivering protein drugs is through the natural secretion machinery of bacteria. In cancer immunotherapy, bacterial toxins, which can penetrate host cells, target various intracellular proteins, and alter host immune responses, are naturally secreted by several bacterial secretory systems. Toxins such as *Vibrio vulnificus* flagellin B, cytolysin A from *S. typhimurium* [44,100], and *Staphylococcus aureus* α-hemolysin have demonstrated a strong capacity to act as adjuvants, eliciting both systemic and mucosal immune responses [122,123]. For example, ClyA, a pore-forming hemolytic protein, kills tumor cells by forming pores in cellular membranes [64,100]. Notably, the antigen-adjuvant fusion protein (tLLO-HPV-16 E7), secreted by the *L. monocytogenes* strain, is currently in a Phase 3 clinical trial for the treatment of cervical cancer (NCT02853604).

Additionally, *Salmonella* and *Yersinia enterocolitica* utilize T3SS injectosomes to inject therapeutic proteins directly into cancer cell cytoplasm in a fast and controllable manner (Figure 2g) [103]. However, for eukaryotic proteins, secretion requires a proper signal peptide, which often presents a bottleneck for recombinant protein drugs. To overcome this challenge, Lynch et al. engineered fully assembled T3SS apparatuses (T3SAs) into some Gram-negative pathogens, enabling the bacteria to function as needle-like extensions that can dock and form pores in the host cell membrane [102]. For example, PROT_3_EcT, an engineered *E. coli* strain with T3SAs, successfully secreted a single-domain antibody and tumor necrosis factor alpha directly into its surroundings (Figure 2h) [102].

## 5. Limitations and Possible Strategies

Bacteria possess several advantageous qualities, including easy genetic manipulation, rapid proliferation, and the ability to specifically target tumor sites, making them a promising living vehicle for treating various diseases, particularly cancer. However, several challenges still need to be addressed, including immune clearance, limited colonization, off-target drug toxicities, and safety concerns.

To overcome these issues, innovative approaches for managing bacteria in therapeutic applications are being actively explored. For instance, to minimize off-target effects, complex genetic circuits with tightly regulated promoters have been developed to regulate the timing of gene expression and payload release, including inducible genetic circuits, quorum sensing systems, and tumor-specific promoters [56]. To improve colonization, intratumoral injection is an effective approach. However, since many tumors are not easily accessible, intravenous injection and oral administration are more practical options for delivering bacteria vaccines. These methods, however, are often associated with challenges such as poor bacteria survival and limited colonization. Several strategies have been proposed to overcome these issues [124]. For instance, vasculature-disrupting agents (VDAs) can significantly enhance bacteria colonization in spontaneous tumors prior to intravenous administration [125]. Additionally, pre-treatment with antibiotics to remove the existing tumor microbiome may create more space for therapeutic bacteria to colonize [126]. Other approaches, such as the use of external magnetic fields and encapsulating bacteria in protective layers (e.g., polysaccharides or apoptotic bodies), have also been shown to help bacteria accumulate at tumor sites [127,128].

One of the primary concerns with using bacteria as therapeutics is the risk of bacterial infection due to their virulence and toxigenic potential, especially in bacteria containing components like lipopolysaccharide (LPS) in the outer membrane of Gram-negative bacteria and other harmful metabolites [129,130]. Consequently, when utilizing bacteria for cancer treatment, the potential adverse effects on host cells must be carefully considered. To address these concerns, much research has focused on modifying bacteria to enhance safety and therapeutic efficacy. Genetic engineering is a widely used approach for adjusting bacterial functions, allowing for the careful modulation of bioactive molecules and harmful factors.

For example, the *msbB* gene, responsible for the synthesis of lipid-A (a key component of LPS), has been targeted to reduce bacterial immunogenicity [131]. An *msbB* mutant in *S. typhimurium* has been engineered to lower immunogenic responses and improve tolerance in clinical trials [73,132]. Similarly, the *msbB* mutation in EcN has shown improved tolerance in animal models [133]. Moreover, attenuated *Salmonella* variants, such as VNP20009, △ppGpp, and A1-R, have demonstrated reduced toxicity. Another promising modification involves a point mutation in *clbP* in EcN, which impairs the synthesis of colibactin without affecting its colonization ability. Colibactin, detected in EcN supernatants, has been associated with DNA cross-linking and mutations, potentially leading to genetic damage in host cells. By inhibiting colibactin synthesis, this modification minimizes the risk of such DNA damage, enhancing the safety profile of EcN as a therapeutic vehicle [134].

In recent years, a novel strategy has gained significant attention from scientists: surface decoration of bacteria with artificial small molecules, polymers, nanoparticles, and even cell membrane camouflage. These approaches aim to adjust the actions of living bacterial therapies and address challenges like immune clearance. One such strategy involves designing surface capsular polysaccharides for EcN to protect the bacteria from immune clearance. The capsule, which is synthesized and released along with the therapeutic payload, is lost afterward, facilitating effective in vivo clearance. This approach significantly enhanced bacterial tolerance—by up to tenfold—and improved anti-tumor efficacy in cancerous mouse models [135].

Cao et al. recently proposed an independent surface nanocoating method using cell and lipid membranes to encase bacteria through physical extrusion and chemical decoration [136,137,138]. These nanocoatings enhance the bioavailability and colonization of bacteria at targeted sites by improving their resistance to immune clearance and harsh external environments. Additionally, membrane enclosures help mitigate in vivo side effects by preventing the release of bacterial toxins and limiting the exposure of surface immunogens. This nanocoating strategy is not limited to EcN. It has been successfully applied to other bacterial vehicles, including *Listeria monocytogenes* and *Porphyromonas gingivalis* [139,140]. Moreover, a variety of cell membranes can be employed for bacterial coating, such as yeast membranes, platelets, red blood cells, macrophages, neutrophils, and even cancer cells [65,141,142,143,144,145].

While therapeutic bacteria hold significant promise for targeting cancer, their effects on non-tumor tissue remains uncertain. Specifically, the preferential growth of bacteria in hypoxic regions could pose potential risks in non-cancerous conditions, particularly in diseases such as inflammatory bowel disease (IBD) or ulcerative colitis (UC). The possibility exists that bacterial therapy could inadvertently exacerbate these conditions. Further research is crucial to fully understand the impact of bacterial therapy on such diseases and to identify strategies to minimize any unintended negative effects. In the context of colorectal cancer (CRC), therapeutic bacteria administered to the gut could impact the existing gut microbiota. On the one hand, these bacteria may compete with the resident microbiota for nutrients or disrupt the microbial balance, potentially leading to dysbiosis. On the other hand, the gut microbiota itself may play a role in enhancing the efficacy of cancer immunotherapies [88,146]. Given that the gut microbiota is not yet fully understood, further research is essential to uncover the complex relationship between the microbiota and foreign bacteria. Understanding how the microbiota influences bacterial colonization, immune responses, and therapeutic outcomes will be critical for optimizing live bacterial therapies not only for CRC but also for other cancers.

## 6. Future Perspectives

Vaccines based on living bacteria not only act as standalone therapies but also serve as effective adjuncts to other treatment modalities. A promising trend is the integration of living bacterial vaccines with various immunotherapies to enhance therapeutic outcomes. For instance, chemotherapy can be combined with *S. typhimurium* VNP20009 to deliver chemotherapy agents like doxorubicin directly into host cells, thereby enhancing the drug’s effectiveness [147]. Similarly, an attenuated live *Listeria monocytogenes* has been paired with radioactive isotopes such as ^188^Rhenium and ^32^P, creating a specialized radioactive *Listeria* strain. This combination significantly inhibits the growth of Panc-02 tumors [77,148].

Additionally, living bacteria can be used to guide targeted therapies toward tumor sites. One innovative approach involves probiotic-guided chimeric antigen receptor T cells (ProCARs), which release synthetic targets and chemokines, improving the recruitment of CAR-T cells to tumor tissue for in situ CAR-mediated lysis [149]. In another example, *DH5α* displaying human DR18 effectively activated mesothelin CAR NK cells, enhancing their trafficking into tumors. The combination of therapies boosting TNF signaling and upregulating NK cell markers further extends NK cell survival within the tumor environment [81].

In summary, although bacteria-based therapies have several limitations, including safety concerns and immune clearance challenges, ongoing advances in genetic engineering, surface decoration, and combination strategies hold great promise. The future of bacteria-based cancer immunotherapy lies in overcoming these hurdles to unlock the full therapeutic potential of bacterial vaccines and other living bacterial therapies.

## 7. Conclusions

Bacteria, with their ability to stimulate the immune system, ease of genetic manipulation, and natural tendency to target tumors, have become a promising platform for delivering therapeutic payloads. To achieve the best therapeutic outcomes, bacteria-based cancer vaccines must be carefully designed. This includes selecting effective drugs, safe bacterial chassis, and engineered synthetic circuits to regulate payload synthesis and release.

Additionally, combining different therapeutic approaches—such as integrating immunological and cytotoxic treatments into a single vaccine or pairing bacterial vaccines with cell therapy—can significantly improve treatment efficacy. As new bacterial strains, control circuits, and immunological molecules continue to be developed, the potential for innovative combination therapies expands.

In conclusion, while there are currently few bacteria-based vaccines available on the market, their immense potential makes them a key element for the future of cancer immunotherapy.

## Figures and Tables

**Figure 1 ijms-26-02056-f001:**
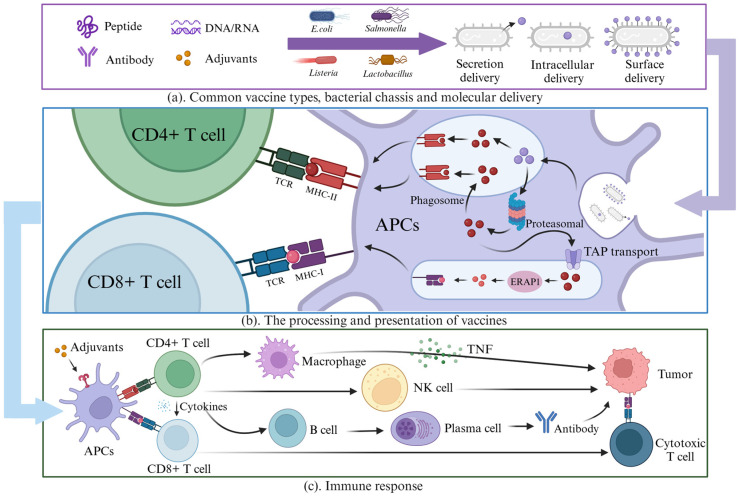
Mechanisms of action in living bacteria-based vaccines. (**a**) Common vaccine types, bacterial chassis, and molecular delivery. (**b**) The processing and presentation of vaccines by APCs and T cells [58]. (**c**) The process of immune response to kill tumor cells. This figure was created with Biorender.com.

**Figure 2 ijms-26-02056-f002:**
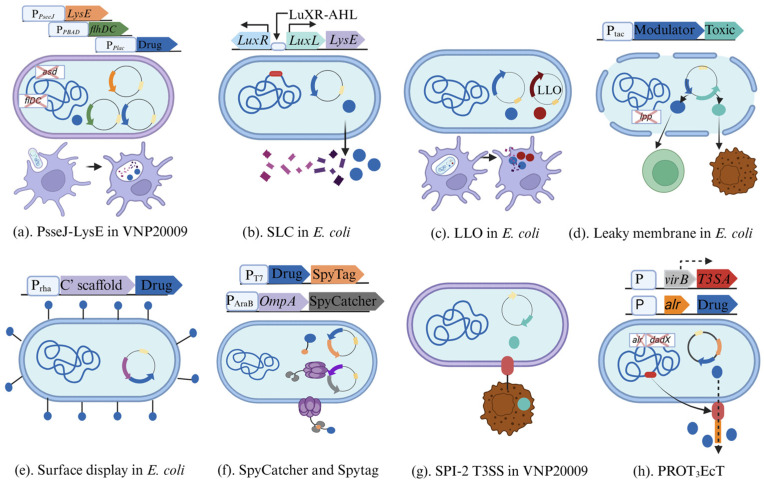
Engineering the bacteria to deliver and release the payload. Engineered bacteria can localize their payload within the bacterial cell (**a**–**d**), on the bacterial surface (**e**,**f**), or secrete it outside the bacteria (**g**,**h**). (**a**) The *PsseJ-lysE* in Salmonella control bacteria lysis and drug release after invasion [91]. (**b**) The SLC system uses the *luxI* promoter and three genes—*AHL*, *luxR*, and *lysE*—to achieve quorum-sensing-based lysis and repeated drug delivery [104,105]. (**c**) LLO creates pores in the bacterial membrane and phagosomal membrane, leading to the relocation of LLO and the drug into the cytosol [106]. (**d**) The leaky bacteria with deletion of the *lpp* gene permits direct release of the payload [97]. (**e**) Surface display of the payload by fusing the gene to an E. coli outer membrane scaffold [84]. (**f**) The SpyCatcher-SpyTag (SC-ST) bioconjugation system is used to surface display the payload, where the drug is fused with ST, and OmpA is fused with SC [107]. (**g**) The native SPI T3SS system in VNP20009 directly injects the payload into tumor cells [103]. (**h**) PROT_3_EcT is engineered by inserting T3SAs into EcT, enabling the release of the payload into the surrounding environment and creating pores in the host cell [102]. This figure was created with Biorender.com.

**Table 1 ijms-26-02056-t001:** Clinical trials of cancer vaccine based on living bacteria.

Species	Payload	Cancer	Phase	Administration	Reference
VNP20009		Advanced solid tumors	Phase I	Intravenous	NCT00006254 ^(1)^
VNP20009		Solid tumors	Phase I	Intravenous	NCT00004216 ^(1)^
VNP20009		Neoplasms	Phase I	Intravenous	NCT00004988 ^(1)^
*Salmonella*	Survivin	Multiple myeloma	Phase I	Oral	NCT03762291 ^(1)^
*Salmonella Typhimurium* SGN1	L-Methioninase	Advanced solid tumors	Phase I	Intratumoral	NCT05038150 ^(1)^
*Salmonella Typhimurium* Saltikva	Interleukin-2	Metastatic pancreatic cancer	Phase II	Oral	NCT04589234 ^(1)^
*BCG*	Monoclonal antibody BEC2	Small cell lung cancer	Phase III	Intradermal	NCT00006352 ^(1)^
*BCG*: S1602		Bladder cancer	Phase III	Intradermal	NCT03091660 ^(1)^
*Clostridium Novyi*-NT Spores		Solid tumors	Phase I	Intratumoral	NCT01924689 ^(1)^
*Clostridium Novyi*-NT Spores	Pembrolizumab	Solid tumors	Phase I	Intratumoral	NCT03435952 ^(1)^
*Listeria*: CRS-207	GVAX vaccine	Metastatic pancreatic cancer	Phase II	Intravenous	NCT01417000 ^(1)^
*Listeria Monocytogenes*	ADXS11-001	Cervical cancer	Phase III	Intravenous	NCT02853604 ^(1)^

^(1)^ ClinicalTrials.gov.

**Table 2 ijms-26-02056-t002:** Examples of bacterial-based delivery strategies in animal models.

Species	Payload	Delivery	Animal Model
*Salmonella*	Caspase-3 [91]	Intracellular lysis by *PsseJ*-*lysE*	Murine breast cancer, liver cancer
*E. coli*	CD47 nanobody [92]	Extracellular lysis by SLC	Lymphoma, melanoma, breast cancer
EcN	IFN-γ [84]	Extracellular lysis by SLC	Colon cancer
EcN	Human CXCL 16 [93]	Extracellular lysis by SLC	Colon cancer, breast cancer
*E. coli* DE3	PD-L1 nanobody and CTLA-4 nanobody [94]	Extracellular lysis by SLC	Lymphoma, colorectal cancer
EcN	Neoepitope array [80]	Intracellular lysis by LLO	Colon cancer, melanoma
*E. coli*	shRNA [95]	Intracellular lysis by LLO	N/A
*Listeria*	TAA(Mage-b) [96]	Intracellular lysis by native LLO	Breast tumor
EcN	Neoleukin-2/15, anti-PDL1 nanobody [97]	Leaky membrane (Δlpp)	Colon cancer
*Lactococcus lactis*	HPV-16 E7 protein [98]	Surface delivery by fusion	Human papillomavirus type 16
*E. coli*	MAP3061c [99]	INP surface display	Mycobacterium Paratuberculosis
*E. coli* DH5α	Decoy-resistant IL18 mutein [81]	Surface delivery by fusion	Colon cancer, melanoma
*Salmonella*	ClyA [100]	Secretion	Pancreatic cancer
*L. monocytogenes*	tLLO-HPV-16 E7 [101]	Secretion	Cervical cancer
PROT_3_EcT	Nanobody [102]	Secretion	N/A
*Staphylococc* *us epidermidis*	Melanoma tumor antigens [87]	Fusion to surface/secretion	Melanoma, prostate cancer

## Data Availability

Not applicable.

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
