# Peer review of "Living Bacteria: A New Vehicle for Vaccine Delivery in Cancer Immunotherapy"

_ijms, 2025, doi:10.3390/ijms26052056_

Round 1

Reviewer 1 Report

Comments and Suggestions for Authors

The authors have provided a comprehensive review that includes mechanisms of action in live bacterial therapy, the types of bacteria used in this therapy, delivery of therapeutic molecules to APC cells and the tumor microenvironment, and limitations and methods for addressing these issues. This will be valuable in understanding recent advances in bacteria-based cancer therapy. Several points are noted.

Possible routes for administering bacteria to cancer patients include intravenous, intra-tumor, oral, inhalation, and intraluminal administration. The advantages and disadvantages of each route should be discussed.

The authors write that bacteria, due to preferentially grow in the hypoxic and necrotic regions of the tumor microenvironment, representing a promising vehicle for delivering a variety of compounds directly to tumors (page 5). Does this mean, for example, that therapeutic bacteria could accumulate not only in cancer but also in ulcerative lesions, a disease of colon? Would this exacerbate colorectal disease? This needs to be addressed in Limitation section.

When colorectal cancer is treated with live bacteria, does the gut microbiota change with treatment? An explanation of how the actual therapeutic effect may be influenced by the existing gut microbiota is needed.

Genetic stability of therapeutic live bacteria is a potential problem (ref. 64). The methods to enhance the instability needs to be explained.

Page 5: BCG, an attenuated strain of Mycobacterium bovis, has long been used clinically to treat bladder cancer. This therapy is effective when the cancer remains in the shallow layers, but not when it has invaded the muscular layer. Does this mean that this is a limitation of immunotherapy in which bacteria are administered into the bladder? The possibility of overcoming the limitations of bacterial therapy for bladder cancer with recently developed methods needs to be explained.

Author Response

Comments 1:

The authors have provided a comprehensive review that includes mechanisms of action in live bacterial therapy, the types of bacteria used in this therapy, delivery of therapeutic molecules to APC cells and the tumor microenvironment, and limitations and methods for addressing these issues. This will be valuable in understanding recent advances in bacteria-based cancer therapy.

Response 1:

We sincerely thank the reviewer for their positive feedback and thoughtful assessment of our review. We appreciate the acknowledgment of the comprehensiveness of the manuscript in addressing the mechanisms of action in live bacterial therapy, the types of bacteria utilized, the delivery of therapeutic molecules, as well as the limitations and methods for overcoming these challenges. We are pleased to know that the review is considered valuable in understanding recent advances in bacteria-based cancer therapy.

We hope that the revisions made have further enhanced the clarity and depth of the manuscript, and we are open to any additional suggestions or requests for improvement.

Comment 2

Possible routes for administering bacteria to cancer patients include intravenous, intra-tumor, oral, inhalation, and intraluminal administration. The advantages and disadvantages of each route should be discussed.

Response 2:

We thank the reviewer for highlighting the importance of discussing the advantages and disadvantages of different administration routes for bacterial therapy in cancer treatment. We appreciate the opportunity to expand on this point.

As noted, in animal models, bacterial vaccines are primarily administered through intratumoral injection. However, in clinical trials (as newly added Table 1 summarized), four main routes of administration are commonly employed: oral, intratumoral, intravenous, and intradermal injection. While there is limited research on bacterial vaccine administration specifically, we have drawn insights from related clinical trials to explore the pros and cons of these routes (referencing Biomaterials-assisted cancer vaccine delivery: preclinical landscape, challenges, and opportunities).

  • Oral administration offers ease of use and avoids the pain associated with injections, but the drug may be subject to digestion and barriers within the gastrointestinal tract and mucus membranes, potentially reducing efficacy.
  • Intratumoral injection allows for localized delivery of bacterial vaccines directly into the tumor site, eliciting a strong local immune response. However, this method is technically demanding and may not be feasible for tumors that are not easily accessible.
  • Intravenous administration facilitates rapid systemic distribution of the bacteria, but this non-targeted delivery can result in the rapid clearance of the bacteria from the body and may lead to potential systemic side effects.
  • Intradermal injection is the most common method for vaccine administration and allows the bacteria to interact with abundant immune cells, enabling sustained release. However, it may also cause potential muscle tissue damage.

In our review, we primarily focus on strategies to enhance bacterial vaccine efficacy, and we briefly mention the different administration routes in the limitations section. We acknowledge that while intratumoral injection is optimal for targeted delivery, it is not always feasible due to accessibility issues. As a result, intravenous and oral routes are more practical, though they may face challenges such as poor bacterial survival and colonization.

We have revised the manuscript to include a more detailed discussion of these administration routes and their respective advantages and limitations, as suggested.

The added text can be found in Limitation and Possible strategies Section, lines 406-410 in track change manuscript:

  • “To improve bacterial colonization, intratumoral injection is an effective approach. However, since most tumors are not easily accessible, intravenous injection and oral administration are more practical methods for delivering bacterial vaccines. These approaches, however, are often associated with challenges such as poor bacterial survival and limited colonization.”

The newly added Table 1 is shown below.

Table 1. Clinical trials of cancer vaccine based on living bacteria.

Species

Payload

Cancer

Phase

Administration

Reference

VNP20009

Advanced Solid tumors

Phase I

Intravenous

NCT000062541)

VNP20009

Solid tumors

Phase I

Intravenous

NCT000042161)

VNP20009

Neoplasm

Phase I

Intravenous

NCT000049881)

Salmonella

Survivin

Multiple myeloma

Phase I

Oral

NCT037622911)

Salmonella Typhimurium SGN1

L-Methioninase

Advanced solid tumor

Phase I

Intratumoral

NCT050381501)

Salmonella Typhimurium Saltikva

Interleukin-2

Metastatic pancreatic cancer

Phase II

Oral

NCT045892341)

BCG

Monoclonal antibody BEC2

Small Cell Lung Cancer

Phase III

Intradermal

NCT000063521)

BCG: S1602

Bladder cancer

Phase III

Intradermal

NCT030916601)

Clostridium Novyi-NT Spores

Solid tumor

Phase I

Intratumoral

NCT019246891)

Clostridium Novyi-NT Spores

Pembrolizumab

Solid Tumors

Phase I

Intratumoral

NCT034359521)

Listeria: CRS-207

GVAX vaccine

Metastatic Pancreatic cancer

Phase II

Intravenous

NCT014170001)

Listeria Monocytogenes

ADXS11-001

Cervical Cancer

Phase III

Intravenous

NCT028536041)

1 ClinicalTrials.gov.

Comment 3:

The authors write that bacteria, due to preferentially grow in the hypoxic and necrotic regions of the tumor microenvironment, representing a promising vehicle for delivering a variety of compounds directly to tumors (page 5). Does this mean, for example, that therapeutic bacteria could accumulate not only in cancer but also in ulcerative lesions, a disease of colon? Would this exacerbate colorectal disease? This needs to be addressed in Limitation section.

Response 3:

We thank the reviewer for raising this important concern. You are correct that bacteria, due to their preference for hypoxic and necrotic regions, could potentially accumulate in not only tumor tissues but also in ulcerative lesions, such as those associated with diseases like inflammatory bowel disease (IBD) or colorectal ulcers.

While the tumor microenvironment (TME) and ulcerative lesions share some similarities, such as low oxygen and high necrosis, it is important to note that the immune landscape and underlying pathophysiology differ significantly. In the case of cancer, bacteria can stimulate an immune response that is beneficial for targeting the tumor. However, in the case of ulcerative lesions, bacterial accumulation might exacerbate inflammation and potentially worsen disease progression.

We have now included this point in the Limitations and Possible Strategies section of the manuscript, clarifying that the preferential growth of bacteria in hypoxic regions may carry potential risks in non-cancerous conditions, especially in diseases like IBD or ulcerative colitis. We have acknowledged that further research is needed to fully understand the impact of bacterial therapy on such conditions and the potential for unintended exacerbation of disease.

We appreciate the reviewer’s insight in highlighting this issue, and the manuscript has been revised accordingly.

The added text can be found in Limitations and Possible strategies Section, lines 459-465 in track change manuscript:

  • “While therapeutic bacteria hold significant promise for targeting cancer, their effects on non-tumor tissues remain uncertain. Specifically, the preferential growth of bacteria in hypoxic regions could pose potential risks in non-cancerous conditions, particularly in diseases such as inflammatory bowel disease (IBD) or ulcerative colitis (UC). The possibility exists that bacterial therapy could inadvertently exacerbate these conditions. Further research is crucial to fully understand the impact of bacterial therapy on such diseases and to identify strategies to minimize any unintended negative effects.”

Comment 4:

When colorectal cancer is treated with live bacteria, does the gut microbiota change with treatment? An explanation of how the actual therapeutic effect may be influenced by the existing gut microbiota is needed.

Response 4:

We thank the reviewer for bringing up this important point regarding the potential impact of live bacterial therapy on the gut microbiota during colorectal cancer (CRC) treatment.

Indeed, the gut microbiota plays a critical role in regulating immune responses and influencing the efficacy of therapies, including those involving live bacteria. The gut is home to a complex microbial community that can interact with therapeutic bacteria, potentially modulating the outcome of the treatment. The composition and diversity of the microbiota may influence how bacteria introduced for therapeutic purposes colonize the gut and how they interact with the host immune system.

In colorectal cancer, live bacteria may not only target the tumor directly but also affect the gut microbiota. For example, therapeutic bacteria might compete with the existing microbiota for nutrients or might alter the microbial balance, potentially leading to dysbiosis. This imbalance in the microbiota could either enhance or reduce the therapeutic efficacy of the bacterial treatment, depending on the nature of the interaction. Additionally, changes in gut microbiota composition could influence immune cell activity, either promoting anti-tumor immunity or contributing to systemic inflammation that might affect the progression of cancer.

We have expanded on this concept in the manuscript, providing a discussion in the Limitations and Possible Strategies section regarding the potential interactions between live bacterial treatments and the gut microbiota. We note that while therapeutic bacteria offer great promise in targeting cancer, their effects on the gut microbiota are not yet fully understood and require further investigation. Understanding how the microbiota influences bacterial colonization, immune responses, and therapeutic outcomes will be essential for optimizing live bacterial therapies for CRC and other cancers.

The added text can be found in Limitations and Possible Strategies Section, lines 465-474 in track change manuscript:

  • “In the context of colorectal cancer (CRC), therapeutic bacteria administered to the gut could impact the existing gut microbiota. On one hand, these bacteria may compete with the resident microbiota for nutrients or disrupt the microbial balance, potentially leading to dysbiosis. On the other hand, the gut microbiota itself may play a role in enhancing the efficacy of cancer immunotherapies [88, 146]. Given that the gut microbiota is not yet fully understood, further research is essential to uncover the complex relationship between the microbiota and foreign bacteria. Understanding how the microbiota influences bacterial colonization, immune responses, and therapeutic outcomes will be critical for optimizing live bacterial therapies not only for CRC but also for other cancers.”

Comment 5:

Genetic stability of therapeutic live bacteria is a potential problem (ref. 64). The methods to enhance the instability needs to be explained.

Response 5:

We thank the reviewer for highlighting the critical issue of genetic stability in therapeutic live bacteria. Common strategies including genetic manipulation and surface decoration not only improve the safety of the bacteria, but also solve many limitations including genetic instability. As mentioned in the manuscript in line 404-406, “complex genetic circuits and tightly regulated promoters have been developed to regulate the timing of gene expression and payload release, including inducible genetic circuits, quorum sensing systems, and tumor-specific promoters [57]”, which could compromise the stability of the bacterial strain. Meanwhile “surface decoration of bacteria with artificial small molecules, polymers, nanoparticles, and even cell membrane camouflage” can also help protect the live bacteria from environmental factors that might cause genetic instability. These methods provide a controlled environment for the bacteria, reducing exposure to stresses that could lead to genetic mutations.

Page 5: BCG, an attenuated strain of Mycobacterium bovis, has long been used clinically to treat bladder cancer. This therapy is effective when the cancer remains in the shallow layers, but not when it has invaded the muscular layer. Does this mean that this is a limitation of immunotherapy in which bacteria are administered into the bladder? The possibility of overcoming the limitations of bacterial therapy for bladder cancer with recently developed methods needs to be explained.

Response:

We thank the reviewer for bringing up this critical point regarding the limitations of Bacillus Calmette-Guérin (BCG) therapy in bladder cancer treatment. As noted, BCG, an attenuated strain of Mycobacterium bovis, has been an established immunotherapy for superficial bladder cancer. However, as the reviewer correctly points out, its efficacy decreases when the cancer has invaded deeper layers of the bladder wall, particularly the muscular layer. This limitation stems from the fact that BCG primarily elicits a localized immune response, and its effectiveness is often dependent on the ability to reach and stimulate immune cells in the tumor site, which becomes more difficult as the tumor penetrates deeper layers. So to design such vaccine, it requires much energy to finely arrange the selection of payload, the bacteria strain, delivery way, genetic controlled payload and good colonization or some novel strategies to enhance the efficacy of bacterial therapy for muscle-invasive bladder cancer. Maybe some combination therapies suit well for these complex cancers. While future research and clinicals trials are needed to evaluate new vaccines on muscle invasive bladder cancer.

The added text can be found in Bacterial Chassis Section, lines 179-187 in track change manuscript:

  • “However, its efficacy decreases when the cancer has invaded deeper layers of the bladder wall, particularly the muscular layer. This limitation stems from the fact that BCG primarily elicits a localized immune response, and its effectiveness is often dependent on the ability to reach and stimulate immune cells in the tumor site, which becomes more difficult as the tumor penetrates deeper layers. As to design such vaccine for muscle-invasive bladder cancer, it requires much energy to finely arrange the selection of payload, the bacteria strain, delivery way, genetic controlled payload and good colonization or some novel strategies to enhance the efficacy of bacterial therapy for muscle-invasive bladder cancer. Maybe some combination therapies suit well for these complex cancers.”

Reviewer 2 Report

Comments and Suggestions for Authors

In the manuscript, Yang et al. provide a comprehensive review of the use of living bacteria for cancer immunotherapy, covering aspects such as mechanisms of action, commonly used bacterial chassis and payload delivery strategies. However, several areas need improvement before the manuscript is ready for publication.

  1. The authors only focus on how bacteria activate the immune system but do not discuss other important mechanisms, such as bacterial colonization of tumors. A discussion of how bacteria selectively target and colonize tumors should be provided.
  2. The authors only discuss a few bacterial species, such as Salmonella typhimurium, coli, and Clostridium butyricum. More bacterial strains should be introduced. Moreover, a summary table including details such as bacterial type, targeted cancer type, animal models or clinical trials, payloads, and key findings should be provided.
  3. Although Figure 2 illustrates major bacterial-based delivery strategies, an additional table summarizing these strategies with more reports using different bacteria should be provided.
  4. An important and relevant study published last year reporting the use of bacteria to induce antitumor responses in melanoma (DOI: 10.1126/science.abp9563) should be discussed in the manuscript.

Author Response

Comment 1:

In the manuscript, Yang et al. provide a comprehensive review of the use of living bacteria for cancer immunotherapy, covering aspects such as mechanisms of action, commonly used bacterial chassis and payload delivery strategies.

Response 1:

We would like to sincerely thank the reviewer for their positive comments on our manuscript. We are pleased to hear that the review is considered comprehensive, covering key aspects such as the mechanisms of action, bacterial chassis, and payload delivery strategies in the context of using living bacteria for cancer immunotherapy.

We hope that the revisions have further strengthened the clarity and depth of these discussions. We are grateful for the reviewer’s time and insightful feedback, which has been invaluable in improving the quality of the manuscript.

Comment 2:

However, several areas need improvement before the manuscript is ready for publication.

The authors only focus on how bacteria activate the immune system but do not discuss other important mechanisms, such as bacterial colonization of tumors. A discussion of how bacteria selectively target and colonize tumors should be provided.

Response 2:

We thank the reviewer for pointing out the importance of bacterial colonization in the context of bacterial cancer therapy. We agree that while our manuscript focused on the immune-activating effects of bacteria, the mechanisms by which bacteria selectively target and colonize tumors are also crucial to understanding their therapeutic potential.

In response to this valuable comment, we have expanded the manuscript to include several strategies on how to improve bacteria colonization in tumor tissues.

  • Vasculature-disrupting agents (VDAs) would significantly improve bacteria colonization in spontaneous tumors before intravenously administration
  • Removing the existing tumor microbiome might leave more space for therapeutic bacteria by the use of antibiotics beforehand
  • External magnetic fields
  • Encapsulation of protective layer including polysaccharides and apoptotic bodies can also help the bacteria to accumulate in tumors

We have included these mechanisms in the revised manuscript and provided additional references to highlight the importance of bacterial targeting and colonization in tumor therapy.

The added text can be found in Limitation and Possible strategies Section, lines 406-417 in track change version:

  • “To improve colonization, intratumoral injection is an effective approach. However, since many tumors are not easily accessible, intravenous injection and orally administration are more practical options for delivering bacteria vaccines. These methods, however, are often associated with challenges such as poor bacteria survival and limited colonization. Several strategies have been proposed to overcome these issues [124]. For instance, vasculature-disrupting agents (VDAs) can significantly enhance bacteria colonization in spontaneous tumors prior to intravenous administration [125]. Additionally, pre-treatment with antibiotics to remove the existing tumor microbiome may create more space for therapeutic bacteria to colonize [126]. Other approaches, such as the use of external magnetic fields and encapsulating bacteria in protective layers (e.g., polysaccharides or apoptotic bodies), have also been shown to help bacteria accumulate at tumor sites [127, 128].

Comment 3:

The authors only discuss a few bacterial species, such as Salmonella typhimurium, coli, and Clostridium butyricum. More bacterial strains should be introduced. Moreover, a summary table including details such as bacterial type, targeted cancer type, animal models or clinical trials, payloads, and key findings should be provided.

Response 3:

We thank the reviewer for this constructive suggestion to broaden the scope of bacterial strains discussed in the manuscript. We agree that while we focused on a few bacterial species, it would be valuable to introduce additional strains to give a more comprehensive overview of the field.

In response, we have expanded the discussion to include additional bacterial species that have been explored for cancer therapy, such as Listeria monocytogenes, Clostridium, Staphylococcus epidermidis. Each of these strains has shown potential in either targeting specific tumor types or enhancing therapeutic effects through engineered payload delivery. These bacteria have unique characteristics that make them suitable for particular applications in cancer immunotherapy, and their inclusion will provide readers with a more diverse perspective on the potential of live bacterial therapies.

Additionally, in line with the reviewer’s suggestion, we have included a summary table (Table 1) that provides a detailed overview of various bacterial strains used in clinical cancer therapy. The table includes information such as:

  • Bacterial species
  • Targeted cancer type(s)
  • Animal models or clinical trials conducted
  • Payload(s) delivered (e.g., cytokines, immune modulators, or toxins)

This table will help readers easily compare and contrast different bacterial strains and their associated therapeutic strategies, highlighting the diversity of approaches in the field.

The added text can be found in Bacterial Chassis Section.

Lines 216-228 in Bacterial Chassis Section in track change version:

  • Listeria monocytogenes primarily affects the gut and spreads through the lymphatic system. In both mouse and human trials, genetically modified Listeria have proven effective as vectors for delivering antigens to tumors[74-76]. By deleting genes such as actA, inlA, and inlB, which prevent intercellular transport and epithelial cell infection, these modified strains become safe while concentrating their delivery to antigen-presenting cells (APCs) [77]. Once internalized by APCs, the native LLO expressed by Listeria helps lyse the bacterial membrane and vesicle membranes, facilitating the translocation of the payload into the cytosol.

Clostridium novyi (C. novyi) is a spore-forming anaerobe that only germinates in low-oxygen environments, such as those found in tumor necrosis or the gut lumen, which keeps healthy tissue intact [78]. Deletion of the alpha toxin makes it a safe therapeutic vehicle [78]. Single injections of non-toxic C. novyi produced immunological responses and showed antitumor activity in an early clinical trial (NCT01924689).”

Lines 258-264 in Bacterial Chassis Section in track change version:

Staphylococcus epidermidis, as part of the skin microbiome, can induce a highly specific adaptive immune response. It has been genetically engineered to express melanoma tumor antigens, leading to the production of tumor-specific T cells that target and kill melanoma cells[87]. Furthermore, intestinal microbes have been shown to generate metabolites that promote carcinogenesis and influence the effectiveness of immunological treatments[88]. Microbiomes present in various tumor sites can both promote and inhibit cancer cell growth [89, 90].”

The newly added Table 1 for bacterial-based delivery strategies in animal models is shown below.

Table 1. Clinical trials of cancer vaccine based on living bacteria.

Species

Payload

Cancer

Phase

Administration

Reference

VNP20009

Advanced Solid tumors

Phase I

Intravenous

NCT000062541)

VNP20009

Solid tumors

Phase I

Intravenous

NCT000042161)

VNP20009

Neoplasm

Phase I

Intravenous

NCT000049881)

Salmonella

Survivin

Multiple myeloma

Phase I

Oral

NCT037622911)

Salmonella Typhimurium SGN1

L-Methioninase

Advanced solid tumor

Phase I

Intratumoral

NCT050381501)

Salmonella Typhimurium Saltikva

Interleukin-2

Metastatic pancreatic cancer

Phase II

Oral

NCT045892341)

BCG

Monoclonal antibody BEC2

Small Cell Lung Cancer

Phase III

Intradermal

NCT000063521)

BCG: S1602

Bladder cancer

Phase III

Intradermal

NCT030916601)

Clostridium Novyi-NT Spores

Solid tumor

Phase I

Intratumoral

NCT019246891)

Clostridium Novyi-NT Spores

Pembrolizumab

Solid Tumors

Phase I

Intratumoral

NCT034359521)

Listeria: CRS-207

GVAX vaccine

Metastatic Pancreatic cancer

Phase II

Intravenous

NCT014170001)

Listeria Monocytogenes

ADXS11-001

Cervical Cancer

Phase III

Intravenous

NCT028536041)

1 ClinicalTrials.gov.

We believe these additions will strengthen the manuscript and provide a more comprehensive view of the current landscape of bacterial therapies in cancer treatment.

Comment 4:

Although Figure 2 illustrates major bacterial-based delivery strategies, an additional table summarizing these strategies with more reports using different bacteria should be provided.

Response 4:

We thank the reviewer for their insightful comment regarding Figure 2 and the suggestion to include an additional table summarizing bacterial-based delivery strategies. We agree that a table would provide a clearer and more concise overview of the various strategies, especially when discussing how different bacterial species can be used to deliver therapeutic payloads.

In response to this suggestion, we have added a new table (Table 2) that summarizes the major bacterial-based delivery strategies, incorporating more examples of different bacteria used in these approaches. The table includes:

  • Bacterial species
  • Therapeutic payload(s) (e.g., cytokines, immune checkpoint inhibitors, toxins)
  • Type of delivery strategy (e.g., direct bacterial colonization, bacterial-induced immune modulation, genetic engineering for payload expression)
  • Specific cancer types targeted

This table complements the existing Figure 2 by providing additional detail and broadening the scope of bacterial-based delivery strategies across various bacterial species. It will help readers better understand the diversity of approaches being investigated in this field and their applications in cancer therapy.

We hope that this addition will enhance the clarity and utility of the manuscript for the readers.

The newly added Table 2 for bacterial-based delivery strategies in animal models is shown below.

Table 2. Examples of bacterial-based delivery strategies in animal models.

Species

Payload

Delivery

Animal model

Salmonella

Caspase-3

Intracellular lysis by PsseJ-lysE

Murine breast cancer, liver cancer

E. coli

CD47 nanobody

Extracellular lysis by SLC

Lymphoma, melanoma, breast cancer

EcN

IFN-γ

Extracellular lysis by SLC

Colon cancer

EcN

Human CXCL 16

Extracellular lysis by SLC

Colon cancer, breast cancer

E. coli DE3

PD-L1 nanobody and CTLA-4 nanobody

Extracellular lysis by SLC

Lymphoma, colorectal cancer

EcN

Neoepitope array

Intracellular lysis by LLO

Colon cancer, melanoma

E. coli

shRNA

Intracellular lysis by LLO

N/A

Listeria

TAA(Mage-b)

Intracellular lysis by native LLO

Breast tumor

EcN

Neoleukin-2/15, anti-PDL1 nanobody

Leaky `membrane (Δlpp)

Colon cancer

Lactococcus lactis

HPV-16 E7 protein

Surface delivery by fusion

Human papillomavirus type 16

E. coli

MAP3061c

INP surface display

Mycobacterium

paratuberculosis

E. coli

Decoy-resistant IL18 mutein

Surface delivery by fusion

Colon cancer, melonoma

Salmonella

ClyA

Secretion

Pancreatic cancer

L. monocytogenes

tLLO-HPV-16 E7

Secretion

Cervical cancer

PROT3EcT

nanobody

Secretion

N/A

Staphylococcus epidermidis

Melanoma tumor antigens

Fusion to surface/secretion

Melanoma, prostate cancer

Comment 5:

An important and relevant study published last year reporting the use of bacteria to induce antitumor responses in melanoma (DOI: 10.1126/science.abp9563) should be discussed in the manuscript

Response 5:

We sincerely thank the reviewer for bringing up the important and relevant study (DOI: 10.1126/science.abp9563)published last year on the use of bacteria to induce antitumor responses in melanoma. We fully agree that this study provides significant insights into the potential of bacteria-based therapies for melanoma, a major area of interest in cancer immunotherapy.

In response to the reviewer’s suggestion, we have incorporated a discussion of this study in the Bacterial Chassis section of the manuscript. Specifically, we highlight how the study demonstrated the use of engineered bacteria to modulate the immune microenvironment in melanoma, resulting in a robust antitumor immune response. The study’s findings have been referenced and discussed in the context of other bacterial-based cancer therapies, providing a more up-to-date view of how recent advances are shaping the field.

We believe that the inclusion of this study further strengthens the manuscript and reflects the latest progress in the field of bacterial immunotherapy for cancer treatment.

The added text can be found in Bacterial Chassis Section, lines 258-261 in track change version:

  • Staphylococcus epidermidis, as part of the skin microbiome, can induce a highly specific adaptive immune response. It has been genetically engineered to express melanoma tumor antigens, leading to the production of tumor-specific T cells that target and kill melanoma cells [87].”

Round 2

Reviewer 2 Report

Comments and Suggestions for Authors

The authors have addressed all my concerns. I agree that the manuscript can be published in its current form. Thanks.